# Amyloid Beta Peptides Lead to Mast Cell Activation in a Novel 3D Hydrogel Model

**DOI:** 10.3390/ijms241512002

**Published:** 2023-07-26

**Authors:** Jingshu Liu, Sihan Liu, Li Zeng, Irene Tsilioni

**Affiliations:** 1Department of Immunology, Tufts University School of Medicine, 136 Harrison Avenue, Boston, MA 02111, USA; jingshu.liu@tufts.edu (J.L.);; 2Program in Cell, Molecular and Developmental Biology, Graduate School of Biomedical Sciences, Tufts University, 136 Harrison Avenue, Boston, MA 02111, USA; 3Program in Pharmacology, Graduate School of Biomedical Sciences, Tufts University, 136 Harrison Avenue, Boston, MA 02111, USA; 4Program in Immunology, Graduate School of Biomedical Sciences, Tufts University, 136 Harrison Avenue, Boston, MA 02111, USA; 5Department of Orthopaedics, Tufts Medical Center, 800 Washington Street, Boston, MA 02111, USA

**Keywords:** Amyloid beta, Alzheimer’s disease, brain, IgE, IL-33, neuroinflammation, mast cells, three-dimensional culture

## Abstract

Alzheimer’s disease (AD) is a prevalent neurodegenerative disease and the world’s primary cause of dementia among the elderly population. The aggregation of toxic amyloid-beta (Aβ) is one of the main pathological hallmarks of the AD brain. Recently, neuroinflammation has been recognized as one of the major features of AD, which involves a network of interactions between immune cells. The mast cell (MC) is an innate immune cell type known to serve as a first responder to pathological changes and crosstalk with microglia and neurons. Although an increased number of mast cells were found near the sites of Aβ deposition, how mast cells are activated in AD is not clear. We developed a 3D culture system to culture MCs and investigated the activation of MCs by Aβ peptides. Because collagen I is the major component of extracellular matrix (ECM) in the brain, we encapsulated human LADR MCs in gels formed by collagen I. We found that 3D-cultured MCs survived and proliferated at the same level as MCs in suspension. Additionally, they can be induced to secrete inflammatory cytokines as well as MC proteases tryptase and chymase by typical MC activators interleukin 33 (IL-33) and IgE/anti-IgE. Culturing with peptides Aβ1-42, Aβ1-40, and Aβ25-35 caused MCs to secrete inflammatory mediators, with Aβ1-42 inducing the maximum level of activation. These data indicate that MCs respond to amyloid deposition to elicit inflammatory responses and demonstrate the validity of collagen gel as a model system to investigate MCs in a 3D environment to understand neuroinflammation in AD.

## 1. Introduction

Alzheimer’s disease (AD) is the most common cause of dementia and disability among the elderly population, affecting an estimated 6.5 million Americans and posing an economic burden of USD 345 billion [1]. One definite sign of AD is the aggregation of toxic amyloid-beta (Aβ) and tau tangles spreading throughout the brain, leading to neuronal death [2]. Emerging evidence suggests that neuroinflammation in AD contributes as much to the pathogenesis as do the plaques and tangles themselves [3]. Multiple types of immune cells were found to be increased in appearance near the sites of Aβ deposition, including microglia, astroglia, T cells, or mast cells (MCs) [4,5,6,7]. These cells interact with each other to provide immune surveillance and contribute to neuroinflammation. 

MCs are long-lived tissue immune cells that are involved in health and disease [8,9]. MC-progenitors can traverse the blood–brain barrier (BBB) and blood–spinal cord barrier under states of inflammation and infection [10]. MCs store large amounts of inflammatory mediators (e.g., TNF-α, histamine, tryptase) in their granules, which can be released within minutes to hours upon activation, enabling MCs to have a fast and strong impact on inflammation despite their relatively small number [11,12,13,14,15,16]. These inflammatory mediators can cause BBB leakage and further exacerbate neuroinflammation [17]. Of note, an isolated brain MC-chymotrypsin-like protease has been implicated in AD, since it has been shown to generate the N-terminus of the Alzheimer Aβ protein [18], ultimately participating in Aβ aggregation deposits. In addition, the exposure of primary microglia, the resident immune cells in the brain, to MC-derived tryptase stimulated microglia to subsequently secrete TNF-α and IL-6 [19]. As a result, MCs instigate and perpetuate the inflammatory activity of microglia and other immune cells in the central nervous system, leading to the progression of AD and other neurodegenerative diseases [20]. 

Recent studies showed an increased number of MCs in the brain of AD patients, especially in areas close to amyloid plaques [6,20]. Mouse and human studies demonstrated that inhibiting MC activation reduced neuroinflammation and led to cognitive benefits, thus providing direct evidence of MCs’ importance in AD [21,22]. Yet, how MCs are activated in the context of AD is hardly clear. 

The objective of our study was to investigate whether Aβ peptides activate MCs to secrete inflammatory mediators in a 3D culture environment that consists of a major extracellular protein, collagen. Prior work studying MCs mostly involved culturing cells in suspension in tissue culture medium [23,24]. However, mature MCs are tissue-resident immune cells surrounded by an extracellular matrix (ECM) in a 3D setting in vivo and so should be studied in 3D. Since collagen I is the major component of ECM in the brain and one of the most used materials for 3D culture development [25,26,27], we chose to encapsulate MCs in the collagen I-formed gel. Although prior studies have cultured MCs in collagen gels in cocultures, none of these have assessed the behaviors of MCs [28,29,30]. Here, we show that MCs cultured in this 3D setting can be activated by interleukin 33 (IL-33) and IgE/anti-IgE signaling to secrete MC-specific proteases and inflammatory cytokines. Furthermore, we report that MCs cultured in collagen gels can be activated by fibrillar Aβ peptides, which could be the basis for serving as a first responder to abnormal protein aggregation in the brain to initiate neuroinflammation in AD.

## 2. Results

### 2.1. MCs Are Present in Mouse Brain Surrounded by Collagen-Rich Extracellular Matrix

To design a 3D cell culture model of MCs that mimics the in vivo environment better, we considered collagen gel. We first confirmed the relevance of collagen to MCs by staining collagen with Van Gieson’s reagent. As expected, collagen is widespread throughout the tissue, as evidenced by the red staining (Figure 1A) [31]. Brain MCs, marked by tryptase, are also present in the brain, embedded in collagen fibers alongside neurons (PGP9.5 positive) (Figure 1B). This result validates the use of collagen gel to culture MCs.

### 2.2. MCs Grown in 3D Collagen Gels Proliferate Similar to MCs Cultured in Suspension

We next cultured MCs in collagen gels to mimic MCs in a 3D environment in the brain. Because there are no established methods to isolate and culture primary brain human MCs, we used LADR MCs. LADR cells were derived from CD34+ cells following the marrow aspiration of a patient with aggressive mastocytosis with no identified mutations in tyrosine-protein kinase (KIT) [32]. It was reported that compared to other MC cell lines, such as LAD2 cells, LADR released higher levels of β-hexosaminidase upon FcεRI crosslinking and a higher level of tryptase. Thus, LADR was considered to resemble primary MCs the most [33]. To facilitate mass transport, we cultured MCs in collagen gels under gentle rotation (60 rpm) (Figure 2A). Since MCs cultured in 3D collagen gels had not been evaluated before, we performed cell viability and proliferation analysis of LADR MCs (Figure 2B,C). Because MCs were traditionally cultured in suspension, this culture method was included in our study for comparison with the 3D method. Both MCs cultured in suspension or in collagen gels were seeded at the same density and analyzed by the Live/Dead assay. Our results showed that MCs cultured in collagen gels were 95% alive, a percentage comparable to those in the suspension culture (Figure 2B). To evaluate whether MCs cultured in collagen gels can proliferate normally, we applied EdU and chased the labeling for 2 h. When cell proliferation was quantified, we found that approximately 3–4% of the cells have incorporated EdU for both MCs in suspension and in 3D collagen gels, suggesting similar proliferation rates (Figure 2C). These results indicate that LADR MCs cultured in 3D collagen gels can survive and proliferate the same way as those in suspension.

### 2.3. MCs Cultured in 3D Collagen Gels Respond to IL-33 and IgE/anti-IgE to Release Inflammatory Mediators

We next determined whether MCs cultured in 3D collagen gels can respond to signals that cause MC activation. Various inflammation-related molecules have been implicated in AD pathogenesis [34]. Among them, IL-33 is highly expressed in various tissues and cells in the mammalian brain and plays a key role in neuroinflammation [35,36]. IL-33 administration in mice led to a spatial memory performance deficit associated with an increase of inflammatory markers in the hippocampus [35]. To survey proteins that MCs can secrete, we performed a human inflammatory cytokine array, which profiles multiple inflammatory cytokines, chemokines related to AD, as well as MMP9 and VEGF [36,37,38] (Figure 3A). We chose a time point of 24 h after IL-33 treatment, which is a time point that yielded the best release of cytokines (Appendix A). Overall, MCs cultured in 3D collagen gels are similarly activated as those cultured in suspension. IL-6 is the most highly elevated cytokine, whose release reached over 1000 folds, while IL-13 and GM-CSF were induced by approximately 839 and 438 folds, respectively (Figure 3A). Other highly induced cytokines are IL-2, IL-5, IL-8, RANTES, and TNF-α, whose induction ranged from 22 to 137 folds (Figure 3A). All these highly induced cytokines have been reported to be elevated in AD [39,40,41,42,43,44,45,46]. To confirm this result, we chose to perform ELISA on a moderately induced cytokine IL-1β. Although the cytokine array showed IL-1β to be induced by 4 folds, ELISA indicated a 25-fold induction by IL-33 (Figure 3A,B). Thus, while the cytokine array analysis was not as sensitive as ELISA, our results demonstrated that MCs in 3D cultures can respond to IL-33 to secrete inflammatory mediators relevant to AD neuroinflammation. 

Two key MC-specific proteases are tryptase and chymase, which are prestored in MC granules and released by MCs within minutes to hours upon stimulation [47]. These enzymes are not included in the cytokine array, so we performed ELISA to evaluate whether MCs cultured in 3D collagen gels can secrete tryptase and chymase. Since IL-33 does not lead to degranulation [48,49,50], we used another trigger, IgE. IgE is critical for allergic responses [51,52,53]. Several studies showed that allergy has been implicated in AD [54,55,56]. Therefore, we first sensitized LADR MCs in suspension and in a 3D collagen gel culture with IgE for 24 h, then stimulated them with anti-IgE for 2 h. Our results showed that MCs cultured in both conditions significantly increased the secretion of tryptase and chymase in the cell culture supernatants, compared to controls (Figure 3C). This result indicates that MCs cultured in 3D can be triggered to degranulate as well, in addition to producing cytokines (Figure 3C). Thus, we consider that MC collagen gel culture can serve as a model to investigate the response of MCs under AD pathogenesis.

### 2.4. Aβ Peptides Stimulate the Release of Cytokines from Human MCs in the 3D Collagen Gel

Amyloid β peptides are a hallmark of AD pathology. Most notable peptides are Aβ1-42, Aβ1-40, and Aβ25-35. Aβ1-42 shows significant abundance in certain forms of AD [57], while Aβ1-40 represents the most abundant Aβ isoform in AD cerebral cortex [58]. Aβ1-42 has two extra hydrophobic amino acids compared to Aβ1-40, which promotes greater fibrillar formation in Aβ1-42 and is known to be more toxic [59]. Aβ25-35, the shortest Aβ fragment, retains some of the amyloidogenic and cytotoxic properties of the other two peptides [60]. No prior studies have compared all three peptides’ effects on human MCs, although two studies have used selected ones to investigate histamine release in rodent MCs cultured in suspension [14,15].

MCs in 3D culture demonstrated a dose-dependent effect for Aβ peptides after 24 h treatment with 10 μM and 50 μM of Aβ1-42, Aβ1-40, and Aβ25-35 in our cytokine array analysis. Interestingly, cytokines that were highly induced by Aβ peptides were not identical to those highly induced by IL-33, suggesting MCs respond to different stimuli differently (Figure 4A). For example, with 50 µM of Aβ1-42 treatment, while GM-CSF and IL-2 were still greatly induced (272 folds, 390 folds, respectively), and IL-8 (98 folds), IL-33 (27 folds), and IL-5 (26 folds) were also induced (Figure 4A). However, other highly induced cytokines were GRO (280 folds), IL-1α (64 folds) (Figure 4A). Among three peptides, MCs responded most to Aβ1-42 at the concentration of 50 µM. While Aβ1-40 and Aβ25-35 are often similarly effective at 10 µM (Figure 4A,B). Of note, Aβ1-40 and Aβ25-35 induced higher levels for some proteins when the lower concentration (10 µM) was used, including IL-5, IL-10, and TNF-α (Figure 4A,B). To further confirm the results, we performed ELISA for IL-1β and TNF-α and found that these two cytokines were indeed induced by Aβ1-42 most, but to a much lesser extent for Aβ1-40. Aβ25-35 did not lead to the production of IL-1β and TNF-α (Figure 4C). These results indicate that Aβ peptides activate MCs with different sensitivities and the 3D collagen gel culture system can be used to study MCs responses. 

## 3. Discussion

Mast cells (MCs) have been implicated in several chronic inflammatory pathologies [61]. Previous observations supported the hypothesis of an involvement of mast cells in Alzheimer’s disease (AD) [6,62,63]. However, there is still no direct evidence for a stimulatory action of Aβ peptides on these cells. Our study constitutes the first study to develop a 3D cell culture model of MCs using collagen hydrogel. We reported that MCs in a 3D gel matrix have similar rates of survival and proliferation compared to MCs grown in suspension. We also showed for the first time that different types of Aβ peptides, including Aβ1-42, Aβ1-40, and Aβ25-35, activate MCs encapsulated in a 3D collagen gel, to secrete numerous proinflammatory cytokines, such as IL-1β, TNF-α, and IL-6. Notably, among all three Aβ peptides, Aβ1-42 could induce the highest level of cytokines. 

Three-dimensional (3D) culture systems have gained increasing interest due to their ability to mimic tissue-like structures more effectively than monolayer cultures [64,65]. MCs function in a 3D tissue environment, and they are an ideal immune cell candidate for studying inflammatory processes in in vitro 3D tissue models. To design a 3D cell culture model of MCs that mimics the in vivo brain milieu better, we chose collagen I gels because collagen I is widely spread in brains and this 3D system is well established for other cell types [25,26,27,66,67]. Several other studies have used collagen to develop 3D culture models for neurons, demonstrating cell survival, proliferation, and differentiation [68,69]. While MCs have been encapsulated in 3D collagen gel models alongside other cell types, such as fibroblasts [28], human airway smooth muscle cells (HASM) [29], as well as chondrocytes [30], no study has assessed the behaviors of MCs and only focused on other cell types in 3D cultures. Thus, our study has overcome the limitations of previous work and laid the foundation for future analysis that recapitulates the 3D environment relevant to AD.

We showed that MCs in the brain were in the same area as the neurons, an area that sees amyloid deposits in AD [70,71,72]. This is consistent with prior reporting indicating MCs and neurons throughout the body both in the peripheral nervous system (PNS) and the central nervous system (CNS) [73]. These close anatomic associations between MCs and neurons are especially evident at sites of inflammation [74,75]. Moreover, MCs express receptors for and are regulated by various neurotransmitters, neuropeptides, and other neuromodulators [76,77,78,79]. Thus, MCs may crosstalk with neurons to substantially influence the behavior and activation of neurons, or other neuronal cells to sustain neuroinflammation and neurodegeneration [80]. 

Based on our cytokine array analysis, we show that IL-33 can induce multiple inflammatory cytokines, as well as MMP9 and VEGF. However, IL-33 is likely to have additional effects when combined with other MC activators. We had previously reported that IL-33 has a synergistic effect with the peptide Substance *p* in stimulating the secretion of IL-1β from human MCs in suspension [77,81], even though it still could not induce degranulation [48]. On the other hand, IgE/anti-IgE could activate degranulation and the release of preformed mediators tryptase and chymase in MCs. These studies support the notion that MCs can be activated in different ways, which could be important in fine-tuning the inflammatory milieu of the AD brain.

One study showed that activated tryptase-positive MCs are in close association with amyloid plaque lesions in brain samples from AD patients [6], raising the possibility of amyloid peptide involvement in the activation of MCs. We showed for the first time that different types of Aβ peptides (Aβ1-42, 1-40, and 25-35) stimulated the release of various cytokines from human MCs in a 3D cell culture model. In the study of Niederhoffer et al., fibrillar Aβ peptides (Aβ1–40 and Aβ1–42) induced the rapid degranulation of histamine from cultured peritoneal-derived MCs [15], and histamine secretion was dependent on the CD47/β1 integrin/Gi protein membrane complex. In another study, Harcha et al. showed that exposure of bone marrow-derived MCs to Aβ25–35 peptide induced the degranulation of histamine via a mechanism that depends on Panx1 HCs [14]. However, these studies did not investigate whether these Aβ peptides can induce the secretion of cytokines or degranulation and release of other important proteases. Our studies provided an important advancement in this area and directly compared the three peptides. We found that Aβ1-42 was the most potent one among all three Aβ peptides, which is consistent with previous observations mentioning that fibrillar amyloid peptide Aβ1-42 is usually considered as the most pathological [82]. On the other hand, other amyloid peptides at the lower concentration could induce the release of multiple cytokines equally well or even better than Aβ1-42, indicating a differing sensitivity that was not uncovered in prior studies. It will be intriguing to investigate the underlying mechanism by which MCs induce inflammatory cytokines upon Aβ treatment. As the secretion of some of the cytokines requires transcription, we expect Aβ has altered the transcriptome of MCs. Others, such as TNF-α, could be released via degranulation. Since we performed the analysis at 24 h post treatment, what we observed could be a combination of de novo synthesis and degranulation. Because of the robust responses to Aβ treatment by MCs, we do not expect cytotoxicity by Aβ at these concentrations at this time point, concentrations that also did not cause toxicity to another type of immune cell microglia [83,84]. However, it will be interesting to investigate the effect of Aβ in longer term cultures.

A limitation of our study is that we used immortalized MCs instead of primary MCs to develop our 3D culture model due to the lack of feasibility to isolate human brain MCs. As primary cells will likely demonstrate better physiological relevance compared to immortalized cells, future studies would involve testing our system in rodent brain MCs. We may also utilize MCs from genetically modified animals. While cells in suspension culture is also a 3D culture system, MCs in suspension culture are not surrounded by ECM, as they are in brain tissues. Since ECM is known to affect cell behavior, it is not surprising that we observed some discrepancies between MCs cultured in collagen gel vs. suspension. For example, MCs cultured in suspension showed more induction of IL-6 and tryptase, while MCs cultured in collagen gels exhibited more significant induction of IFN-γ and MCP-1. By investigating how MCs behave in collagen gels or gels containing other ECM molecules, we may begin to understand MCs in their natural milieu in the brain. 

In summary, in this study, we demonstrate that collagen gel is a promising approach for the development of a 3D culture model of MCs, since it supports MCs activation by conventional triggers and mimics the brain tissue environment. Our studies also show that different Aβ peptides stimulate 3D gel-cultured MCs to secrete various inflammatory mediators, thus contributing to the understanding of the mechanism of AD neuroinflammation.

## 4. Materials and Methods

### 4.1. Tissue Preparation

All animal care and experimental procedures were approved by the Institutional Animal Care and Use Committee (IACUC) at Tufts University. Two-month-old C57BL/6 male mice (Jackson Lab, Boston, MA, USA) were caged in groups under standard conditions before being euthanized. Brains were isolated according to a previously published protocol [85] and then fixed in 10% formalin for paraffin sectioning. Brain sections were deparaffinized and rehydrated with xylene and gradient ethanol before staining. 

### 4.2. Human Mast Cell Culture

LADR mast cells, kindly supplied by Dr. A. S. Kirshenbaum, National Institutes of Health, Bethesda, MD, USA [33], were cultured in StemPro-34 serum free medium (Invitrogen, Waltham, MA, USA, #10639011) with nutrient supplement, 2mM glutamine, 100 U/mL penicillin/streptomycin and 100 ng/mL recombinant human stem cell factor (rhSCF, Stemgen, Milan, Italy), kindly supplied by Swedish Orphan Biovitrum AB (Stockholm, Sweden). Cells were maintained at 37 °C in a humidified incubator in an atmosphere of 95% O_2_/5% CO_2_. Hemi-depletions were performed weekly. 

### 4.3. 3D Collagen Gel Culture Preparation

Collagen gel mixture was prepared by mixing 68% complete StemPro media, 30% collagen type I (Corning, Corning, NY, USA, #354236), 2–2.5% 0.8M NaHCO_3_ (Sigma-Aldrich, St. Louis, MO, USA, #S5761) [86,87]. LADR cells were centrifuged and resuspended in the collagen gel mixture at a density of 2.5 × 10^5^ cells/mL. A total of 10 µL volume beads were seeded into 24-well tissue culture plates for mast cell treatments, 25 µL volume beads were seeded into 4-well chamber slides for the EdU cell proliferation assay and the Live/Dead assay. A total of 500 μL medium was added into each well. 

### 4.4. Mast Cell Treatments

For IgE/anti-IgE treatment, cells were sensitized with IgE (1 μg/mL; Sigma-Aldrich, St. Louis, MO, USA, #AG30P) overnight and then triggered with anti-IgE (2 μg/mL; Invitrogen, Waltham, MA, USA, #H15700) for 2 h. For IL-33 treatment, IL-33 (100 ng/mL; R&D Systems, #3625-IL) was applied in the medium. For Aβ peptide treatment, solutions of Aβ1-40, Aβ1-42, and Aβ25-35 were incubated at 37 °C for 72 h to fibrillate [15]. Fibrillar Aβ peptides 1–42 (10 μM and 50 μM), 1–40 (10 μM and 50 μM) and 25–35 (10 μM and 50 μM) (AnaSpec, Fremont, CA, USA, #AS-20276, #AS-24235, #AS-24227) were then applied to MCs for 24 h.

### 4.5. ELISA and Cytokine Array Analysis

Supernatants were collected and subjected to ELISA for IL-1β, Chymase (R&D Systems, Minneapolis, MN, USA, #DY201-05 and #DY4099-05) and Tryptase (BosterBio, Pleasanton, CA, USA, #EK0898) and Human Cytokine Array Q1 (RayBiotech, Peachtree Corners, GA, USA; #QAH-CYT-1), according to manufacturer’s suggestions. The OD values of samples for ELISA were read at 450 nm using a SpectraMax M5 plate reader (Molecular Devices, San Jose, CA, USA). The cytokine array slides were visualized by the laser scanner at Cy3 wavelength. The Q-Analyzer, an excel-based tool developed by the manufacturer, was used for cytokine array data analysis. Each antibody is arrayed in quadruplicate on the cytokine array slides. Heatmaps were plotted using the log2 value of the fold change of each protein after treatments, using GraphPad Prism 9. Three-dimensional bar charts were generated in Excel using the fold change of each protein after treatment compared with controls.

### 4.6. EdU Cell Proliferation Assay

EdU cell proliferation assay was performed using Click-iT™ Plus EdU Cell Proliferation Kit for Imaging (Invitrogen, Waltham, MA, USA, #C10637). Cells were seeded into 4-well chamber slides then treated with 10 µM 5-ethynyl-2′-deoxyuridine (EdU) for 2 h. Cells cultured in collagen gels were directly examined under the microscope for the percentage of EdU positive cells. Cells cultured in suspension were encapsulated in collagen gel to assess the percentage of EdU positive cells. The results were quantified by ImageJ.

### 4.7. Live/Dead Assay

The Live/Dead assay was performed using LIVE/DEAD™ Viability/Cytotoxicity Kit (Invitrogen, Waltham, MA, USA, #L3224) according to the manufacturer’s protocol. The results were quantified using MATLAB, as described previously [88].

### 4.8. Histological and Immunofluorescence (IF) Staining

Van Gieson’s staining solution, i.e., 0.1% picro-fuchsin solution, was used for assessing the collagen. It was made by mixing 1% acid fuchsin aqueous solution (acid fuchsin powder: Thermo Scientific, Waltham, MA, USA, #400210250) with 1.2% picric acid aqueous solution (RICCA Chemical, Arlington, TX, USA, #5860-16) at 1:9 ratio. Weigert′s iron hematoxylin solution (Sigma-Aldrich, St. Louis, MO, USA, #HT1079-1SET) was used to stain nuclei. For immunofluorescence staining, antigen retrieval was performed with 10mM citrate buffer with 0.05% Tween-20, pH 6.0. Sections were stained with the following primary antibodies: UCHL1/PGP9.5 Polyclonal antibody (1:300, Proteintech, Rosemont, IL, USA, #14730-1-AP, gift from Dr. Brian Lin, Tufts University) and tryptase antibody (1:100, Abcam, Waltham, MA, USA, #ab151757). Secondary antibodies are the Alexa Fluor 488-conjugated goat anti-rabbit secondary antibody (Invitrogen #A-11008) and the Alexa Fluor 594-conjugated goat anti-rabbit secondary antibody (Invitrogen, Waltham, MA, USA, #A-11012). DAPI was used to counterstain sections. Sections incubated with only one primary antibody served as single-positive controls. Sections incubated with secondary antibodies only served as negative controls. 

### 4.9. Microscopy

Bright field and fluorescent images were taken using an Olympus IX-71 microscope and Olympus DP80 digital camera (Olympus, Waltham, MA, USA), with the Olympus cellSens software standard 1.18.

### 4.10. Statistical Analysis

All in vitro conditions were performed in triplicate and all experiments were repeated at least three times. The results are presented as mean ± standard error of the mean (SEM). For the cytokine array, each antibody was spotted in quadruplicate in each array. Differences between the two groups were assessed using Student’s *t*-test. Comparisons among at least three groups were tested using one-way analysis of variance (ANOVA), and then post hoc comparisons to determine the significant differences between several experimental groups and the control group and between two groups were performed using Dunnett’s test and the Bonferroni test, respectively. Differences with *p*-values less than 0.05 were considered statistically significant. Shapiro–Wilk normality tests were performed to assess the normality of the data. Datasets with *p*-values higher than 0.05 were considered as following the normal distribution (Appendix A). All analyses were performed using Graph Pad Prism 9. 

## Figures and Tables

**Figure 1 ijms-24-12002-f001:**
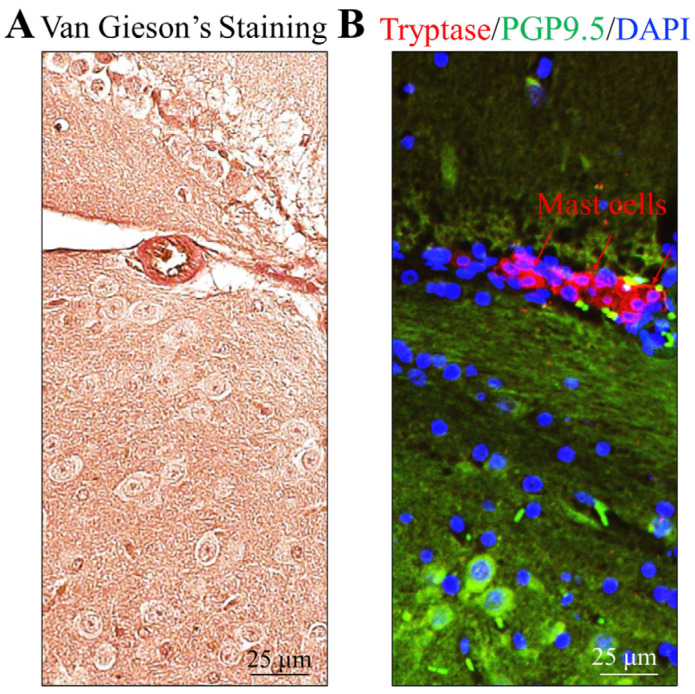
**MCs are embedded in collagen fibers in the brain.** (**A**). Van Gieson’s staining was used on mouse brain sections to show the presence of collagen fibers (red). Weigert′s iron hematoxylin was used to stain nuclei. (**B**). Tryptase and PGP9.5 double immunofluorescence was performed to localize MCs (Tryptase positive, red, arrows) and neurons (PGP9.5 positive, green arrows). DAPI (blue) was used for staining nuclei.

**Figure 2 ijms-24-12002-f002:**
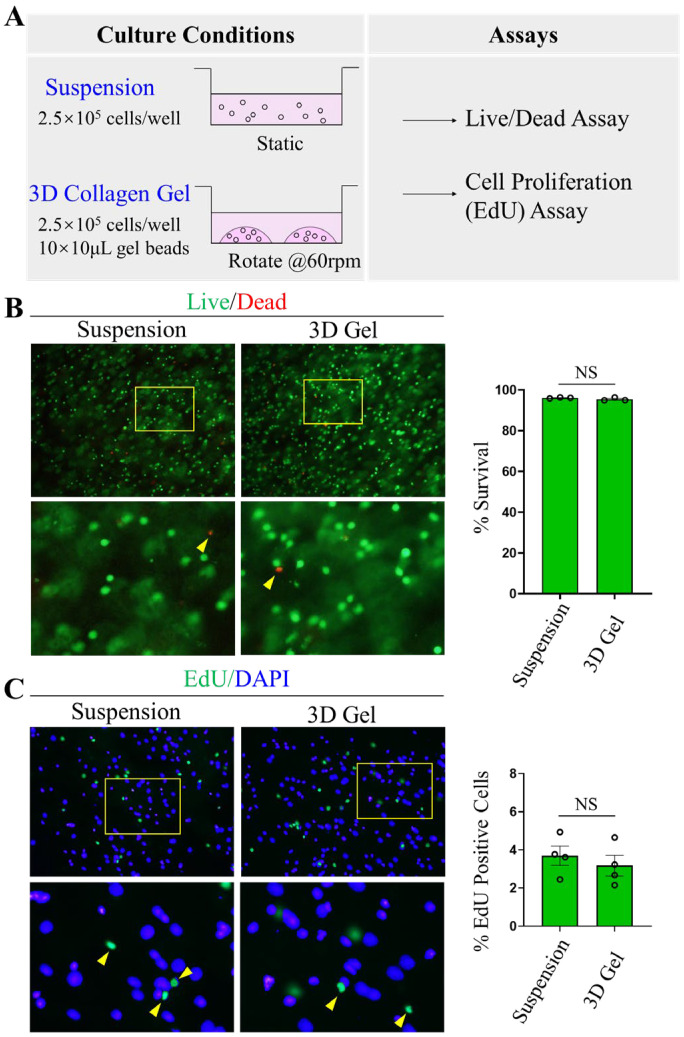
**MCs are viable and proliferative in 3D collagen gel cultures.** (**A**). The schematic of the 3D culture of MCs in collagen gels. (**B**). Cell survival after 24 h of culturing as measured by Live/Dead assay. Live cells: green; dead cells: red. The percentage of live cells were quantified. The experiment was performed three times independently. Results are shown as the representative experiment. Top row: images for Live/Dead analysis, 20× magnification. Yellow boxes show the areas magnified as the bottom row images. Bottom row: dead cells (red) are indicated by yellow arrows. Only focused cells were counted for analysis. (**C**). Cell proliferation assessment using the EdU cell proliferation assay. EdU was introduced into the medium for 2 h and the percentage of cells incorporated with EdU was quantified. The experiment was performed three times independently. Results are shown as the representative experiment. Top row: images for EdU analysis, 20× magnification. Yellow boxes show the areas magnified as the bottom row images. Bottom row: green cells are EdU positive and indicated by yellow arrows. Only focused cells were counted for analysis.

**Figure 3 ijms-24-12002-f003:**
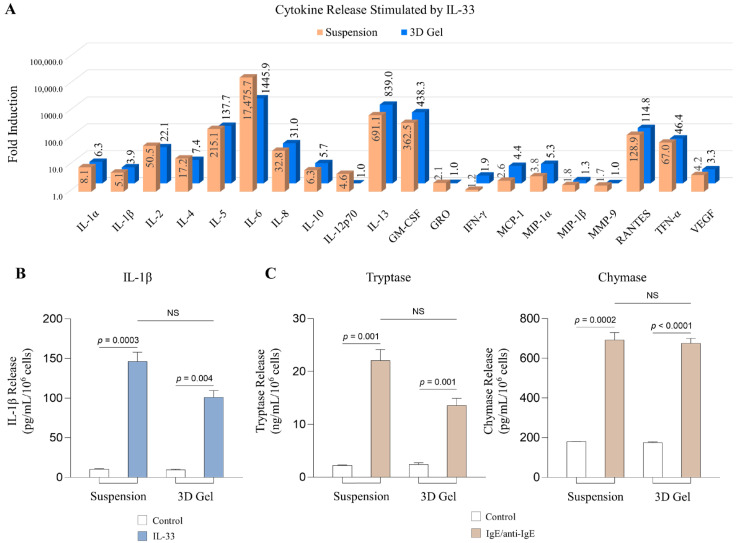
**MCs cultured in 3D collagen gels respond to IL-33 and IgE/anti-IgE.** (**A**). Cytokine analysis was performed on supernatants of MCs cultured in suspension or in 3D collagen gels and treated with IL-33 (100 ng/mL) for 24 h, using the Human Inflammatory Cytokine Array Kit (Raybiotech). Each treatment includes four biological repeats and four technical repeats. The release of each protein after IL-33 treatment compared to control was quantified as fold induction. (**B**). The release of IL-1β was further confirmed by ELISA in MCs treated with IL-33 for 24 h. NS: The difference between compared groups is not significant. (**C**). MCs were first sensitized by IgE (1 µg/mL) for 24 h, then stimulated by anti-IgE (2 µg/mL) for 2 h. Tryptase and chymase release from MCs after IgE/anti-IgE treatment was assessed by ELISA. Each experiment was performed three times independently. Results are shown as the representative experiment. NS: The difference between compared groups is not significant.

**Figure 4 ijms-24-12002-f004:**
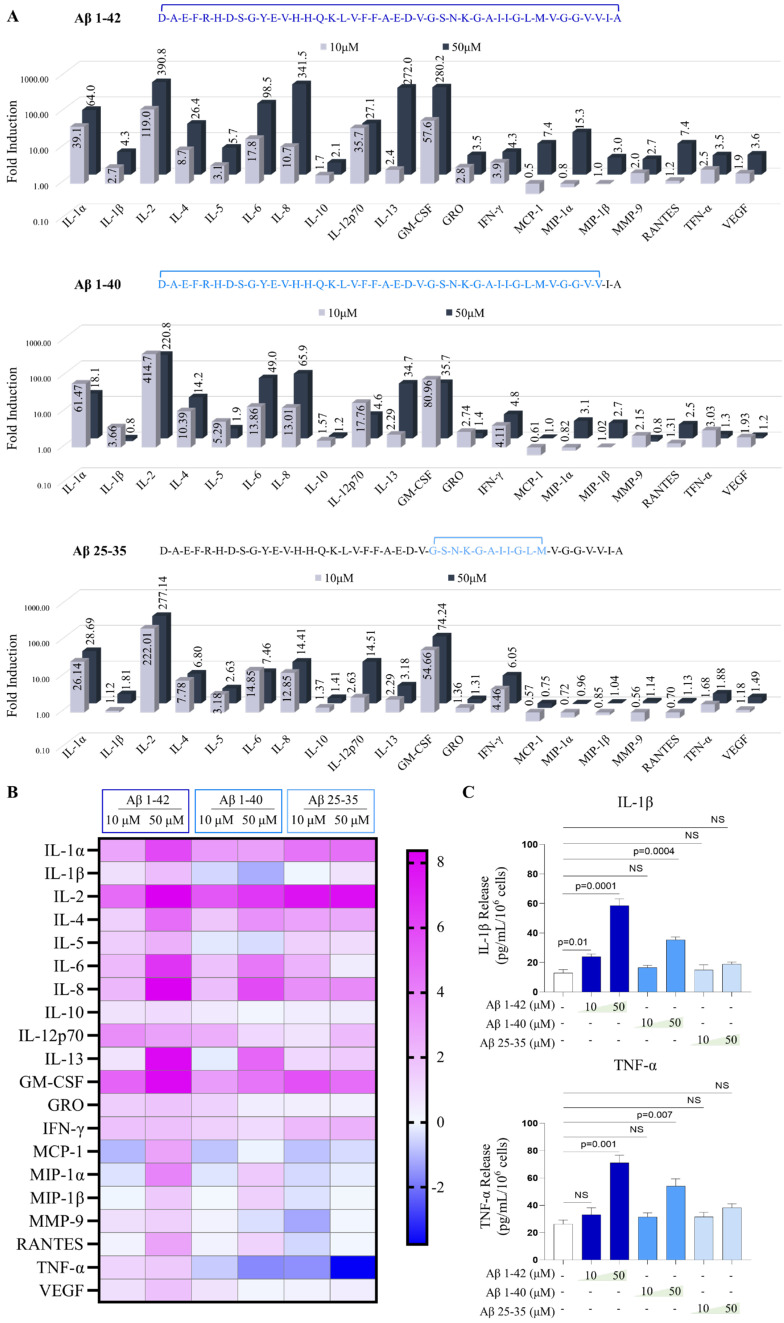
**MCs in 3D collagen gel culture can be activated by Amyloid-β peptides to different extents.** (**A**). Cytokines secreted from MCs in 3D collagen gels induced by three Amyloid-β peptides for 24 h: Aβ1-42, Aβ1-40, and Aβ25-35 were shown as fold induction. (**B**). Heatmap showing the comparison of cytokine release induced by all three Aβ peptides. (**C**). Amyloid-β peptides stimulated IL-1β and TNF-α release which was measured by ELISA. Experiments were independently performed three times. Results are shown as the representative experiment.

## Data Availability

Available upon request.

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
