# Peer review of "Amyloid Beta Peptides Lead to Mast Cell Activation in a Novel 3D Hydrogel Model"

_ijms, 2023, doi:10.3390/ijms241512002_

Round 1

Reviewer 1 Report

In this manuscript titled “Amyloid Beta Peptides Lead to Mast Cell Activation in a Novel 3D Hydrogel Model” the authors demonstrated the etiology of amyloid beta peptides induced pathology through mast cell activation. In that case, they encapsulated mast cells in the collagen 3D hydrogel. The authors gave enough evidence to use this collagen hydrogel as a mimic of the physiological environment to match up with the in vivo observation. After establishing it as a standard, the authors showed that Aβ1-42 is more harmful to cells compared to Aβ1-40 and Aβ25-35 as the previous one activates the MCs at maximum levels. The manuscript was well written but, still in my opinion, the following suggestions will help this manuscript more suitable for diverse readers.

1)      The font sizes and their types have some problems throughout the manuscript. Authors may take care of it.

2)      On page 2, the authors mentioned “we report that MCs cultured in collagen gels respond differently to different fibrillar Aβ peptides, which could be the basis for serving as a first responder to abnormal protein aggregation in the brain to initiate neuroinflammation in Alzheimer’s Disease.” As authors explained rest of the part more elaborately in the introduction part, they reframe this sentence by explaining the “respond differently” to maintain consistency.

3)      Figure 3A is not readable. Authors may work on it to improve its resolution.

4)      In the Figure 3A bar graph, there are few releases that are comparable to gel and suspension MCs but there are few releases that are not comparable or rather opposite. It may be included in the limitation section of this work.

5)      In Figures 3B and C, in comparison to control, both suspension and gel showed several-fold increments but their release was not comparable to each other. In the discussion, the author may discuss a little bit more about the difference between suspension and gel release of IL-1β and tryptase.     

6)       Figure 4A has some resolution issues.

7)      On page 7, the authors mentioned about “we performed ELISA for IL-1β and TNF-α and found that these two cytokines were indeed induced by Aβ1-42, but to a much lesser extent for Aβ1-40. Aβ1-40 did not lead to the production of IL-1β and TNF-α (Fig. 4C).” figure 4C showed different trends that were stated in the abovementioned statement. I think that the authors wanted to mean Aβ25-35 instead of Aβ1-40 in the last sentence of the statement.

8)      On page 10, the authors mentioned: “Several other studies have used collagen to develop 3D culture models for neurons, demonstrating cell survival, proliferation, and differentiation66,6”. At the same time, the authors were claiming that they are the first person to do this kind of hydrogel. Authors may put more emphasis on the uniqueness that they are bringing.

9)      On page 10, the authors also mentioned: “Moreover, MCs express receptors for and are regulated by various neurotransmitters, neuropeptides, and other neuromodulators77-80”. It means that the differences are associated with these components. Authors may explain it more elaborately in the context of the differences or similarities between suspension and gel.

Author Response

Manuscript ijms-2480991

Response to Reviewers

Dear Editor,

Thank you for giving us the opportunity to submit a revised draft of the manuscript “Amyloid Beta Peptides Lead to Mast Cell Activation in a Novel 3D Hydrogel Model” for publication in the International Journal of Molecular Sciences. We appreciate the time and effort that you and the reviewers dedicated to providing feedback on our manuscript and are grateful for the insightful comments on and valuable improvements to our paper. We have incorporated most of the suggestions made by the reviewers. Those changes are highlighted within the manuscript. Please see below, in blue, for a point-by-point response to the reviewers’ comments and concerns.

______________________________________________________________________________

Reviewer 1

Comment #1: The font sizes and their types have some problems throughout the manuscript. Authors may take care of it.

Author Response: Thank you for pointing this out. We suspect that this is due to the formatting upon submission, as the font in our submitted manuscript was uniform. Nevertheless, we apologize for this and have made the correction accordingly. We hope the editor would allow us to submit a pdf file as well for your review.

Comment #2: On page 2, the authors mentioned “we report that MCs cultured in collagen gels respond differently to different fibrillar Aβ peptides, which could be the basis for serving as a first responder to abnormal protein aggregation in the brain to initiate neuroinflammation in Alzheimer’s Disease.” As authors explained the rest of the part more elaborately in the introduction part, they should reframe this sentence by explaining the “respond differently” to maintain consistency.

Author Response: Thank you for your comment. We agree that we should make this sentence consistent with our prior induction. We have now changed the sentence on page 2 to: “We report that MCs cultured in collagen gels can be activated by fibrillar Aβ peptides, which could be the basis for serving as a first responder to abnormal protein aggregation in the brain to initiate neuroinflammation in Alzheimer’s Disease”.

Comment #3: Figure 3A is not readable. Authors may work on it to improve its resolution.

Author Response: We agree with this comment. This subfigure illustrated what the raw signal looked like in our cytokine array studies. Each slide was 6cm x 2cm in size, with all the antibodies printed on it, including the standard curve and 4 technical repeats/antibody. This is why each dot was extremely small. These slides were then scanned for quantitative analysis, which we showed in the subsequent raw figure. Considering Fig. 3A merely illustrated the raw signal prior to scanning and all results were shown in Fig. 3B, we have decided to remove the image from the manuscript and to show only the results from the analysis.

Comment #4: In the Figure 3A bar graph, there are few releases that are comparable to gel and suspension MCs but there are few releases that are not comparable or rather opposite. It may be included in the limitation section of this work.

Author Response: Thank you for your comment. Yes, a few releases indeed show differences between MCs in the gel and suspension. For example, MCs cultured in suspension rather than in collagen gel showed IL33 induction of IL-6 and IL-12p70. On the other hand, MCs cultured in collagen gel showed a more significant induction of IFN-γ and MCP-1. These differences could be due to how cells respond to their environment, in suspension of culture medium vs 3D collagen. Since the MCs in the brain are surrounded by extracellular matrix that comprises of collagen and other molecules, and extracellular matrix is known to affect cell behavior, investigating how MCs behave in collagen gel may help us understand MCs in their natural milieu in the brain. We have included this in the discussion.

Comment #5: In Figures 3B and C, in comparison to control, both suspension and gel showed several-fold increments, but their release was not comparable to each other. In the discussion, the author may discuss a little bit more about the difference between suspension and gel release of IL-1β and tryptase.

Author Response: Thank you for your comment. Mature MCs are naturally present in a 3D environment, which regulates gene expression. Thus, it is expected that cells cultured in suspension and 3D gel could have slightly different protein expression upon IL-33 treatment. In Figures 3B and 3C, MCs cultured in 3D seemed to release lower levels of IL-1β and tryptase than those cultured in suspension. However, such differences are not statistically significant. We have discussed this in the revised version of the manuscript.

Comment #6: Figure 4A has some resolution issues.

Author Response: Thank you. We apologize for this. We believe this was caused when the manuscript was formatted. In the revision, we inserted Figure 4A to resolve the resolution issues. We hope we would be allowed to submit a pdf version of the manuscript to provide best clarity.

Comment #7: On page 7, the authors mentioned “we performed ELISA for IL-1β and TNF-α and found that these two cytokines were indeed induced by Aβ1-42, but to a much lesser extent for Aβ1-40. Aβ1-40 did not lead to the production of IL-1β and TNF-α (Fig. 4C).” figure 4C showed different trends that were stated in the abovementioned statement. I think that the authors wanted to mean Aβ25-35 instead of Aβ1-40 in the last sentence of the statement.

Author Response: Thank you for this comment. You are correct that we meant Aβ25-35 instead of Aβ1-40 in the last sentence of the statement. We apologize for this oversight. We have revised the manuscript accordingly.

Comment #8: On page 10, the authors mentioned: “Several other studies have used collagen to develop 3D culture models for neurons, demonstrating cell survival, proliferation, and differentiation66,67”. At the same time, the authors were claiming that they are the first person to do this kind of hydrogel. Authors may put more emphasis on the uniqueness that they are bringing.

Author Response: Thank you for this comment. We have clarified this statement. Indeed, collagen gels have been used by other investigators, but our study is the first study to use collagen gel to specifically study the behavior of mast cells. We have revised the manuscript accordingly to emphasize the uniqueness of our study.

Comment #9:  On page 10, the authors also mentioned: “Moreover, MCs express receptors for and are regulated by various neurotransmitters, neuropeptides, and other neuromodulators77-80”. It means that the differences are associated with these components. Authors may explain it more elaborately in the context of the differences or similarities between suspension and gel.

Author Response: This is a good point, since cells cultured in 3D collagen matrix could have slightly altered protein expression as those in suspension, as mature MCs are naturally present in a 3D environment surrounded by extracellular matrix, which regulates gene expression. This might explain some differences in the cytokines we analyzed. We have revised the manuscript to address this important issue.

­­­­­­­­­­­­­­­­­­­­­­­_____________________________________________________________________

Reviewer 2 Report

This study describes the inflammatory effect of Ab peptides on inflammation in mastocytes, and demonstrates the validity of a collagen gel 3D culture model. Therefore, the finding of MCs responding to amyloid deposition is interesting regarding the study of neuroinflammation in Alzheimer’s disease. However, a mechanistic approach of this inflammatory effect is lacking.

My major concerns are the following:

-          Owing to the molecular nature of the journal, it would have been of great interest to address the mechanism behind the Ab peptide-induced inflammation (at transcriptional level or not, induction of degranulation).

-          Did Ab peptides stimulate the release of enzymes by degranulation? Did they induce cytotoxicity? These issues should be discussed.

-          Was a time-course conducted regarding the release of inflammatory cytokines or enzymes in response to the stimuli?

-          I suppose that Figure 4 shows the results obtained with the fibrillated form of the peptides at the conditions described in methods (72h at 37ºC). Was checked the effect of the peptides with several aggregation states? (early aggregation, oligomers, fibrils generated with several times of shaking)

-          The statistical test used assumes a normal distribution of data; however, the number of experiments is too low.

Other comments

-          Figure legends should show the time of incubation with the stimuli and the number of experiments. The resolution of Fig4A is not high enough.

-          To study the effect of secretome from MC on microglia (or other cell types) would have been interesting.

-          In the first paragraph of Introduction there is different size of the letters. I hink that in line 59 “plagues” is incorrect.

Author Response

Manuscript ijms-2480991

Response to Reviewers

Dear Editor,

Thank you for giving us the opportunity to submit a revised draft of the manuscript “Amyloid Beta Peptides Lead to Mast Cell Activation in a Novel 3D Hydrogel Model” for publication in the International Journal of Molecular Sciences. We appreciate the time and effort that you and the reviewers dedicated to providing feedback on our manuscript and are grateful for the insightful comments on and valuable improvements to our paper. We have incorporated most of the suggestions made by the reviewers. Those changes are highlighted within the manuscript. Please see below, in blue, for a point-by-point response to the reviewers’ comments and concerns.

Reviewer 2

Comment #1: Owing to the molecular nature of the journal, it would have been of great interest to address the mechanism behind the Ab peptide-induced inflammation (at transcriptional level or not, induction of degranulation).

Author Response: Thank you for the comment. We have addressed the potential mechanism in the discussion. In our study, we found Aβ-peptide induced the secretion of multiple proteins. Since it has been established that many of these cytokines, such as IL-1β, require de novo synthesis of proteins, we expect that they were induced at the transcription level, so we analyzed them 24hrs after treatment. On the other hand, tryptase and chymase are known to be released via degranulation, which is why we analyzed them 2 hrs after treatment. We have included this discussion in our revised manuscript. In the meantime, we intend to conduct a thorough RNAseq investigation to assess transcriptome changes after Aβ treatment.

“Colin K. Combs, J. Colleen Karlo, Shih-Chu Kao, and Gary E. Landreth. β-Amyloid Stimulation of Microglia and Monocytes Results in TNFα-Dependent Expression of Inducible Nitric Oxide Synthase and Neuronal Apoptosis. J Neurosci. 2001 Feb 15; 21(4): 1179–1188. doi: 10.1523/JNEUROSCI.21-04-01179.2001. PMCID: PMC6762255, PMID: 11160388.”

“Douglas G Walker, John Link, Lih-Fen Lue, Jessica E Dalsing-Hernandez, Barry E Boyes. Gene expression changes by amyloid beta peptide-stimulated human postmortem brain microglia identify activation of multiple inflammatory processes. J Leukoc Biol 2006 Mar;79(3):596-610.doi: 10.1189/jlb.0705377. PMID: 16365156 DOI: 10.1189/jlb.0705377”

Comment#2: Did Ab peptides stimulate the release of enzymes by degranulation? Did they induce cytotoxicity? These issues should be discussed.

Author Response: This are important points. Yes, tryptase and chymase are known to be released via degranulation. The fact we observed these enzymes within 2 hrs of stimulation by Aβ peptides confirms that. While we did not directly test the cytotoxicity by Aβ peptides, we expect that they had not resulted cell death within the 24 hrs period because MCs were able to induce the expression of many inflammatory stimulators. In prior studies on other cell types, Aβ peptides at these concentrations also have not demonstrated cell toxicity:

  1. Combs CK, Karlo JC, Kao SC, Landreth beta-Amyloid stimulation of microglia and monocytes results in TNFalpha-dependent expression of inducible nitric oxide synthase and neuronal apoptosis. J Neurosci. 2001 Feb 15;21(4):1179-88. doi: 10.1523/JNEUROSCI.21-04-01179.2001. PMID: 11160388, PMCID: PMC6762255 DOI: 10.1523/JNEUROSCI.21-04-01179.2001.

  1. Sanz JM, Chiozzi P, Ferrari D et. al. Activation of Microglia by Amyloid β Requires P2X7Receptor Expression J Immunol (2009) 182 (7): 4378–4385.

However, even though there was no over cell death, it does not mean no mild cytotoxicity. We will perform a thorough investigation of cytotoxicity of different peptides and at different concentrations in our future analysis. We have discussed these issues in our revised manuscript.

Comment #3: Was a time-course conducted regarding the release of inflammatory cytokines or enzymes in response to the stimuli?

Author Response: Thank you for this comment. A time course is indeed important regarding cytokine release. In our studies on MCs, we have determined that 24 hrs after stimulation is the best time point, rather than 48hrs. We have included this data in Supplemental Figure 1 in the revised manuscript.

Comment #4: I suppose that Figure 4 shows the results obtained with the fibrillated form of the peptides at the conditions described in methods (72h at 37ºC). Was checked the effect of the peptides with several aggregation states? (Early aggregation, oligomers, fibrils generated with several times of shaking)

Author Response: This is very interesting. We did not check the several aggregation points in this study. We chose this time point and temperature based published studies, in which different time points were tested:

  1. Niederhoffer N, Levy R, Sick E, et al. Amyloid beta peptides trigger CD47-dependent mast cell secretory and phagocytic responses. Int J Immunopathol Pharmacol 2009;22(2):473-83. DOI:10.1177/039463200902200224.
  2. Harcha PA, Vargas A, Yi C, Koulakoff AA, Giaume C, Saez JC. Hemichannels Are Required for Amyloid beta-Peptide-Induced Degranulation and Are Activated in Brain Mast Cells of APPswe/PS1dE9 Mice. J Neurosci 2015;35(25):9526-38. DOI: 10.1523/JNEUROSCI.3686-14.2015.

However, it would be intriguing to test this in the future, since peptides at different aggregation states may induce different levels of inflammation.

Comment #5: The statistical test used assumes a normal distribution of data; however, the number of experiments is too low.

Author Response: Thank you for this comment. Yes, we agree that a lower number is hard for analysis of normal distribution. Thus, we did perform the Shapiro-Wilk normality test in Prism and the data passed the normality test. We also made the normal QQ plot in GraphPad Prism to visualize that the predicted distribution of actual data is very close to the normal distribution. Here is an example using the ELISA results of IL-33 stimulated IL-1β release. Based on our analysis, we concluded that the distribution was normal. However, in the future, we will continue performing normalization tests to ensure the rigor of our analysis. The distribution analysis has now been included in the revised manuscript as Supplemental Figure 2. Below is an example of such normality distribution analysis. X-axis indicates the actual amount of cytokine levels in the IL-33 treatment experiment from ELISA, and the Y-axis indicates what we predict the data to be if the distribution is normal. The result shows very good alignment of the actual level and predicted level, indicating normal distribution.

Comment #6: Figure legends should show the time of incubation with the stimuli and the number of experiments. The resolution of Fig4A is not high enough.

Author Response: We have now included the time of incubation with the stimuli and the number of experiments on Figure Legends. Thank you!

Comment #7: To study the effect of secretome from MC on microglia (or other cell types) would have been interesting.

Author Response: This is an interesting point, and it is in our future plan to study the effect of MC-secretome on microglia, because MCs and microglia are in the vicinity of each other in the brain and their interaction will likely contribute significantly to AD pathology.

Comment #8: In the first paragraph of Introduction there are different sizes of the letters. I think that in line 59 “plagues” is incorrect.

Author Response: Thank you for pointing this out. Yes, we also became aware of this issue when the formatted manuscript was sent to us alongside reviewers’ comments. We suspect this took place upon editorial formatting. This correction has been made accordingly in the revised manuscript. We hope to submit a pdf file as well to ensure the reviewer can review the manuscript in the correct font/size as well as the highest resolution of the figures.

_____________________________________________________________________

Reviewer 3 Report

The research article aims to examine whether 3D culture will affect the proliferation and survival of mast cells and how mast cells are activated by amyloid beta peptides in a 3D collagen gel.

 This research article is well-organized, and the flow of the content is logical and easy to follow. I have a few suggestions that may improve the manuscript as follows:

1.     Unify the font size and font style in the manuscript. There are some typos and grammar errors in the text, please proofread and correct them.

2.     Adjust the font size in Figures for consistency. In Figure 4, the image resolution of 4A is too poor to read. Please replace it with high resolution image or enlarge it.

3.     Please include the hydrogel model in the introduction section and explain why this is a novel 3D model in 2.1.

4.     Why used LADR human mast cell line not LAD2 mast cell line?

5.     Define “KIT” in Line 94.

6.     The definition for IL-33 in Line 119 is not appropriate. Please provide the definition when it is first used.

7.     In Result section 2.3, the description in text does not correspond to or does not match to what illustrated in Figure 3, i.e. Figure 3B in Line 128, 129 and 143.

8.     In line 174, the statement that “MCs responds to Aβ1-40 the most among three peptides”. Please list the supporting evidence and explain why.

9.     In Figure 4C, the ELISA results of IL-1β and TNF-α stimulated by Aβ1-40 at 10 uM and 50 uM are not consistent with the levels shown in the heatmap in Figure 4B. Please explain.  

10.  In M&M section 3.2, please provide the concentration of glutamine used for cell culture.

11.  In M&M section 3.3, correct typos in the subtitle. In line 206, if the authors want to add reference here, please covert the two PMID numbers accordingly.

12.  Both “IgE/anti-IgE” and “IgE/α-IgE” were shown in the manuscript. Please unify the format.

13.  In Line 92, the author’s statement that “Because there are no established methods to isolate and culture primary human MCs” is Arbitrary. There was a protocol (PMID: 20814942) published in 2010.

The manuscript needs to be carefully proofread and re-checked for typos, grammar, syntax, format coherence, etc. 

Author Response

Manuscript ijms-2480991

Response to Reviewers

Dear Editor,

Thank you for giving us the opportunity to submit a revised draft of the manuscript “Amyloid Beta Peptides Lead to Mast Cell Activation in a Novel 3D Hydrogel Model” for publication in the International Journal of Molecular Sciences. We appreciate the time and effort that you and the reviewers dedicated to providing feedback on our manuscript and are grateful for the insightful comments on and valuable improvements to our paper. We have incorporated most of the suggestions made by the reviewers. Those changes are highlighted within the manuscript. Please see below, in blue, for a point-by-point response to the reviewers’ comments and concerns.

Reviewer 3

Comment #1: Unify the font size and font style in the manuscript. There are some typos and grammar errors in the text, please proofread and correct them.

Author Response: We agree with this comment. This point was raised by reviewer 1 and 2 as well (see above). We have made the corrections accordingly.

Comment #2: Adjust the font size in Figures for consistency. In Figure 4, the image resolution of 4A is too poor to read. Please replace it with high resolution image or enlarge it.

Author Response: You are right. This point was raised by reviewer 1 as well (comment 6). We apologize for this. We believe this was caused when the manuscript was type printed. In the revision, we inserted Figure 4A to resolve the resolution issues. We hope we would be allowed to submit a pdf version of the manuscript to provide the best clarity. We have also provided a new Figure 4 with a higher resolution.

Comment #3: Please include the hydrogel model in the introduction section and explain why this is a novel 3D model in 2.1.

Author Response: Thank you. We have now included more information regarding the hydrogel model in the introduction section.

Comment #4: Why used LADR human mast cell line not LAD2 mast cell line?

Author Response: We used LADR human mast cell line because this is a more mature cell line compared to LAD2 (Kirsenbaum et al 2019). They have higher FcεRI/CD117 expression and can be induced to secrete much higher levels of tryptase than LAD2. Thus, LADR is considered superior to LAD2 in recapitulating primary mast cells. We have provided this rationale in the revised manuscript. Thank you for raising this point.

“Kirshenbaum AS, Yin Y, Sundstrom JB, Bandara G, Metcalfe DD. Description and Characterization of a Novel Human Mast Cell Line for Scientific Study. Int J Mol Sci 2019;20(22). DOI: 10.3390/ijms20225520.”

Comment #5: Define “KIT” in Line 94.

Author Response: Thank you. KIT is the receptor tyrosine kinase protein, known as Tyrosine Protein Kinase. We have now defined “KIT” in Line 94 of the manuscript.

“Yarden Y, Kuang WJ, Yang-Feng T et. al. Human proto-oncogene c-kit: a new cell surface receptor tyrosine kinase for an unidentified ligand. EMBO J. 1987 Nov; 6(11): 3341–3351.PMCID: PMC553789, PMID: 2448137.”

Comment #6: The definition for IL-33 in Line 119 is not appropriate. Please provide the definition when it is first used.

Author Response: We agree with this comment. We defined IL-33 when it was first used.

Comment #7:  In Result section 2.3, the description in text does not correspond to or does not match to what illustrated in Figure 3, i.e., Figure 3B in Line 128, 129 and 143.

Author Response: You are right. We apologize for this oversight. We have changed the manuscript accordingly to correspond to what is illustrated in Figure 2. Specifically, it should be Figure 3A for Line 128 and 129, and Fig 3C for Line 143.

Comment #8:  In line 174, the statement that “MCs responds to Aβ1-40 the most among three peptides”. Please list the supporting evidence and explain why.

Author Response: Thank you for this suggestion. We have now changed the statement in line 174 as follows “MCs responds to Aβ1-42 the most among three peptides”. We apologize for the typo.

Comment #9: In Figure 4C, the ELISA results of IL-1β and TNF-α stimulated by Aβ1-40 at 10 uM and 50 uM are not consistent with the levels shown in the heatmap in Figure 4B. Please explain.  

Author Response: You are correct that Aβ1-40 at 10 µM and 50 µM are not consistent with the levels shown in the heatmap in Figure 4B. We are not clear as to why, and we are planning to redo the cytokine array and ELISA in the future. However, we did observe consistency for Aβ1-42 and Aβ25-35.

Comment #10: In M&M section 3.2, please provide the concentration of glutamine used for cell culture.

Author Response: Thank you for this comment. The concentration of glutamine used for cell culture is now added in M&M section 3.2.

Comment #11: In M&M section 3.3, correct typos in the subtitle. In line 206, if the authors want to add reference here, please covert the two PMID numbers accordingly.

Author Response: Thank you for this comment. We are not clear why the two references were reverted to PMIDs. It could be due to the transfer of our Word document to the Journal’s format upon journal formatting. Sorry for the inconvenience. We have now converted the two PMID numbers accordingly.

Comment #12: Both “IgE/anti-IgE” and “IgE/α-IgE” were shown in the manuscript. Please unify the format.

Author Response: As suggested by the reviewer, we have now used the term “IgE/anti-IgE” within the manuscript.

Comment #13: In Line 92, the author’s statement that “Because there are no established methods to isolate and culture primary human MCs” is Arbitrary. There was a protocol (PMID: 20814942) published in 2010.

Author Response: We apologize for not being clear. What we meant was that there were no established methods to isolate and culture primary brain human MCs. We have revised the manuscript accordingly. Thank you!

Round 2

Reviewer 2 Report

I think that all the queries have been properly answered and now the manuscript is acceptable for publication 

Author Response

We would like to thank Reviewer 2 for his constructive feedback.